# Modulation and Regulation of Canonical Transient Receptor Potential 3 (TRPC3) Channels

**DOI:** 10.3390/cells12182215

**Published:** 2023-09-05

**Authors:** Bethan A. Cole, Esther B. E. Becker

**Affiliations:** 1Nuffield Department of Clinical Neurosciences, University of Oxford, Oxford OX3 9DU, UK; 2Kavli Institute for Nanoscience Discovery, University of Oxford, Oxford OX1 3QU, UK

**Keywords:** TRP channel, TRPC3 gating, ion channel pharmacology

## Abstract

Canonical transient receptor potential 3 (TRPC3) channel is a non-selective cation permeable channel that plays an essential role in calcium signalling. TRPC3 is highly expressed in the brain and also found in endocrine tissues and smooth muscle cells. The channel is activated directly by binding of diacylglycerol downstream of G-protein coupled receptor activation. In addition, TRPC3 is regulated by endogenous factors including Ca^2+^ ions, other endogenous lipids, and interacting proteins. The molecular and structural mechanisms underlying activation and regulation of TRPC3 are incompletely understood. Recently, several high-resolution cryogenic electron microscopy structures of TRPC3 and the closely related channel TRPC6 have been resolved in different functional states and in the presence of modulators, coupled with mutagenesis studies and electrophysiological characterisation. Here, we review the recent literature which has advanced our understanding of the complex mechanisms underlying modulation of TRPC3 by both endogenous and exogenous factors. TRPC3 plays an important role in Ca^2+^ homeostasis and entry into cells throughout the body, and both pathological variants and downstream dysregulation of TRPC3 channels have been associated with a number of diseases. As such, TRPC3 may be a valuable therapeutic target, and understanding its regulatory mechanisms will aid future development of pharmacological modulators of the channel.

## 1. Introduction

Transient Receptor Potential (TRP) proteins are a superfamily of ion channel-forming subunits encoded for by 28 genes in mammals. The “canonical” or “classical” transient receptor potential (TRPC) subfamily consists of seven members, TRPC1-7, and has the closest sequence homology to the *Drosophila* TRP channel which was cloned originally (reviewed in [1]). The mammalian TRPC channels are further divided into two subgroups based on their sequence homology and electrophysiological properties: TRPC3/6/7 and TRPC1/4/5. TRPC3 was the first TRPC channel to be heterologously expressed and functionally characterised in vitro and is the most highly expressed TRPC channel subunit in the brain [2,3].

### 1.1. Structure and Function of TRPC3 Channels

Structurally, TRP channels are related to the K_v_ family of voltage-gated K^+^ channels. Subunits are composed of six membrane-spanning helices, a pore-forming re-entrant loop that contains a selectivity filter, and cytoplasmic N- and C-termini (Figure 1A,B). Although the transmembrane domains have no voltage sensor, transmembrane domains S1–4 are often referred to as the voltage sensor-like domains (VSLD). Unlike K_v_ channels, however, TRP channel subunits most commonly form non-selective mono- or divalent cation permeable channels with very modest voltage sensitivity.

Different TRPC channel subunits can assemble to form both functional homotetramers and heterotetramers, which differ in their biophysical properties [4,5,6,7]. TRPC subunits can heteromerise with other subgroup members [8], as well as members of the wider TRP channel family such as TRPV4 and TRPP2 [9,10,11]. Whereas TRPC1 is unable to function as a homomeric channel [4,8,12], TRPC3 and related TRPC6 are often found as homotetrameric channels in physiological settings. 

TRPC3 has a long S3 helix compared to other TRP channels, which forms a negatively charged extracellular cavity [13]. This is a feature shared with TRPC6 [14] and is potentially a site for channel modulation (Figure 1A). The intracellular domain (ICD) of the TRPC3 homotetrameric assembly is formed by four N-terminal ankyrin repeat domains (AR1-4), a highly conserved cytoplasmic TRP helix which connects the ICD to the transmembrane S6 helix via a linker, a cytoplasmic horizontal helix (HH) and vertical helix (VH) [13,14,15,16] (Figure 1).
Figure 1Structural architecture of TRPC3. (**A**) Schematic diagram of TRPC3 subunit topology with key modulation sites highlighted. Helices are represented by cylinders. The negatively charged extracellular cavity (EC) formed by the extended S3 is indicated. Inhibitor site 1 indicates the binding site for SAR-7334 and AM-1473; inhibitor site 2 indicates the binding site for BTDM. The binding site for the activator GSK1702934A (GSK_170_) is shown in yellow. Disease-causing gain-of-function mutations are indicated with a green diamond. (**B**) Highlighted ribbon structure of a single TRPC3 subunit with helices coloured as in (**A**), PDB code 6CUD. Lipid binding sites L1 and L2 with a resolved lipid density (orange) are indicated. The location of Ca^2+^ binding sites (CBS1,2,3) identified by Guo and colleagues [15] (PDB code 7DXB) in TRPC3 are overlaid (cyan spheres). CBS1 is located on the VH, CBS2 on the linker between the HH and VH, and CBS3 is located within the VSLD. *AR* ankyrin repeat domain, *CBS* calcium binding site, *ICD* intracellular domain, *HH* horizontal helix, *LHD* linker helical domain, *NGS* N-linked glycosylation site, *PM* plasma membrane, *TRP* transient receptor potential, *VH* vertical helix.
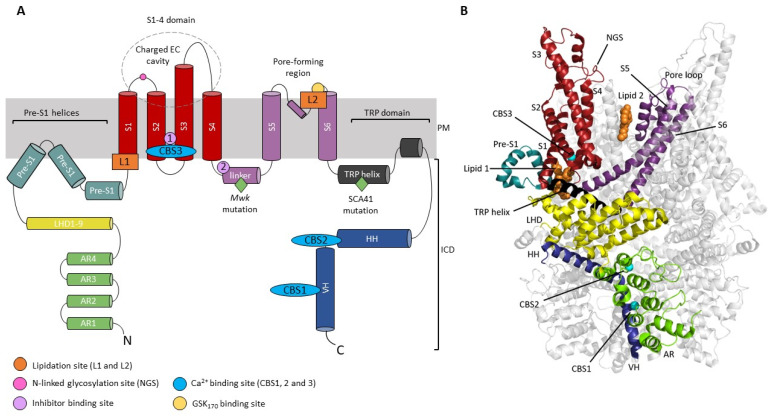


The homotetrameric TRPC3 channel displays only modest selectivity for Ca^2+^ ions over Na^+^ (P_Ca_/P_Na_ = 1.6) [17] and exhibits a higher degree of basal activity than the closely related TRPC6/7 channels. The slight divalent cation selectivity of TRPC3 arises due to a negatively charged glutamate residue in its selectivity filter, E630 [18,19] (UniProt isoform 3/Q13507-3). TRPC3/6/7 are activated downstream of G-protein coupled receptor (GPCR) activation by diacylglycerol (DAG), a product of phospholipase C (PLC) activation, in a membrane-delimited manner [20]. DAG binds directly to the channel, predominantly at a lipidation site termed L2, which is located in the pore-forming region of the channel [21,22,23,24]. In contrast to TRPC6, TRPC3 exhibits high basal activity, which is attributed to mono N-glycosylation at an asparagine residue located on the extracellular loop between the S1 and S2 helices [13,25] (Figure 1), where TRPC6 has dual N-glycosylation [25].

When TRPC3 is heterologously expressed in HEK293 cells, single channel openings are short in duration [20], reflecting a more energetically favourable closed conformation. Currents recorded in the whole-cell configuration usually display a distinct double rectification shape, or a flattening of the current–voltage (I–V) curve at membrane voltages close to the reversal potential and at strongly negative voltages [25]. This feature is shared with TRPC6 and 7 channels, but the mechanisms underlying the complex I-V relationship are currently unknown [26]. Additionally, double rectification is not seen in all heterologous expression systems, so it is unclear if this is an intrinsic property of the channels or whether it arises from endogenous modulators present in certain cell types.

### 1.2. Isoforms of TRPC3

There are three distinct isoforms or splice variants of TRPC3 that have been identified thus far; TRPC3a, TRPC3b and TRPC3c which possess distinct electrophysiological properties and expression patterns. TRPC3b is the most well-characterised of the isoforms. No functional characterisation of TRPC3a has been carried out to date. The shortest isoform is TRPC3c due to lack of exon 9, which encodes a 28-amino acid region spanning approximately half of the CIRB (calmodulin and inositol 1,4,5-trisphosphate (IP3) receptor binding) domain [27]. The CIRB domain is involved in inhibitory modulation of channel activity by inositol triphosphate receptor (IP3R) and calmodulin (CaM) binding [28,29,30]. TRPC3c is highly and selectively expressed in the brain and was shown to exhibit enhanced activity [16,27,31]. It has been proposed that the increased channel open probability of TRPC3c arises as a result of impaired inhibition by CaM [27], although this has not been demonstrated experimentally. A recent structural and electrophysiological study has shed more light onto the possible gating differences between the TRPC3b and TRPC3c isoforms.

The full-length structure of TRPC3b in a closed conformation was resolved by cryogenic electron microscopy (cryo-EM) [16] and, similar to all other TRPC3 structures, the S6 helix of the channel is located immediately adjacent to a cytoplasmic TRP helix, which is connected to the HH via an unresolved loop, termed the C-terminal loop. It was hypothesised that the HH is allosterically coupled to the transmembrane domains of the channel. Upward movements in the cytoplasmic HHs are predicted to be transduced to the transmembrane S6 domain which contains the channel gate, via the TRP helix and C-terminal loop [16]. In agreement with this model, some TRPC3 structures show upwards bending of the horizontal helix to be almost parallel to the TRP helix [13,15] (Figure 2). Conversely, the HHs resolved in other closed or inactive channel structures lie virtually flat, with a 172° angle between HHs from opposite subunits [14,15,16] (Figure 2). Whilst both groups of structures are closed, there are clearly distinct conformational states which may represent different stages of channel activation. 

The authors created a TRPC3c channel construct by introducing a 28-amino acid deletion in TRPC3b and used electrophysiology to functionally characterise its behaviour [16]. Three of the missing residues in TRPC3c are part of the HH in TRPC3b, and 18% of the C-terminal loop is missing compared to full-length TRPC3b. As expected, the channel was more constitutively active than TRPC3b and also more sensitive to pharmacological activation by GSK1702934A (GSK_170_). In order to rule out impaired CaM inhibition as a mechanism underlying the higher constitutive activity, a construct with a six-amino acid deletion was engineered, to shorten the C-terminal loop whilst not impairing the CIRB domain. This channel also displayed increased activity in whole-cell patch clamp experiments, but to a lesser degree than the full 28-amino acid deletion. In contrast, lengthening the C-terminal domain by both four and eight amino acids resulted in lower channel activity compared to wildtype. Together, these results imply that shortening the C-terminal loop, which likely results in upwards movement of the HH, leads to enhanced TRPC3 channel activity by increasing allosteric coupling between the HH and the transmembrane domains.

## 2. The Role of TRPC3 in Normal Physiology and Human Disease

TRPC3 plays an important role in Ca^2+^ homeostasis and entry into cells throughout the body, and both pathological variants and downstream dysregulation of TRPC3 channels have been associated with a number of diseases [32,33,34,35]. As such, the channel may be a valuable therapeutic target. Central nervous system (CNS) diseases and in particular cerebellar ataxia form the main point of focus in this review given genetic evidence from both human patients and mouse models, whereas other pathologies have only been reported in mouse models or cell lines to date.

### 2.1. The Role of TRPC3 in the Nervous System

TRPC channels are widely expressed in the CNS, and TRPC3 is particularly highly expressed in the cerebellum [36]. TRPC3 expression increases in the rat cerebellum from postnatal day 1 to adulthood, whilst most other TRPC channels are strongly downregulated over this time period [37]. TRPC3 is the predominant TRPC channel in Purkinje cells, which are the sole output neuron of the cerebellar cortex [36,37]. TRPC3 is also expressed in glutamatergic type II unipolar brush cells (UBCs) in the cerebellum [38]. In rodents, the short and more active TRPC3c is the most abundantly expressed isoform in the cerebellum [27]. Similarly, in the human brain the ratio of TRPC3c:TRPC3b subunits is significantly higher in the cerebellum compared to other brain regions [31]. There is no information, however, about the relative expression of splice variants in individual cell types in the brain, and high variation in isoform expression was found in the cerebellum between individuals [31].

TRPC3 is essential for metabotropic type 1 glutamate (mGluR1) receptor signalling in the cerebellum (Figure 3). In Purkinje cells, mGluR1 activation evokes a slow inward excitatory postsynaptic current (sEPSC). This sEPSC is mediated entirely by TRPC3 channels as mGluR1 currents recorded from Purkinje cells in acute cerebellar slices are absent in *Trpc3*−/− mice [36]. Additionally, the functional association between mGluR1 and TRPC3 is thought to be important for the induction of long-term depression in the cerebellum [39]. TRPC3-mediated Ca^2+^ entry has been linked to control of IP3R activity at endoplasmic reticulum–plasma membrane junctions in HEK293T cells to regulate Ca^2+^ homeostasis [40]. IP3R undergoes biphasic regulation by Ca^2+^ ions and elevations in cytosolic Ca^2+^ mediated by TRPC3 inhibit IP3R activity [40]. Given the importance of IP3R signalling in Purkinje cells [41] and its activation downstream of mGlur1 (Figure 3), it is conceivable that TRPC3 may regulate IP3R function in Purkinje cells, but this remains to be shown experimentally.

In addition to being essential for mGluR1 signalling, TRPC3 is involved in the heterogeneous spontaneous firing of Purkinje cells by acting as a pacemaking channel in distinct subpopulations of Purkinje cells. The TRPC3 expression pattern in Purkinje cells is somewhat complementary to zebrin, where the highest expression of TRPC3 is found in the faster firing, zebrin-negative (Z-) Purkinje cells. In these cells, TRPC3 contributes to spontaneous firing and action potential firing frequency in response to current injections [42]. This is likely due to its leak-type conducting properties and high levels of constitutive activity at negative voltages. Selective knockout of *Trpc3* from Purkinje cells results in a decrease in spontaneous firing frequency in Z- but not Z+ Purkinje cells [42]. Similarly, pharmacological inhibition of TRPC3 reduces spontaneous firing frequency in Z- cells [43].

As a result of the essential role for TRPC3 in the cerebellum, dysfunction of the channel has been implicated in cerebellar disease. Gain-of-function point mutations in TRPC3 have been shown to cause spinocerebellar ataxia (SCA), both in a patient with SCA41 and the Moonwalker (*Mwk*) mouse mutant [44,45]. In the *Mwk* mouse, mutation of a threonine to alanine (T635A) leads to constitutive activation of TRPC3 [38,45,46]. The mouse mutant has a severely ataxic gait and shows impaired Purkinje cell dendritic arborization and eventual cell death [45]. Type II UBCs are also completely ablated in the *Mwk* mouse [38]. Whilst the exact mechanism by which the *Mwk* mutation results in constitutive TRPC3 activity is unclear, the mutation is located in the S4–S5 linker (Figure 1A). This region is thought to be important for channel gating and indeed, antagonists which limit flexibility of this linker impair gating [13,47], which will be discussed later. Interestingly, the human SCA41 TRPC3 mutation (R762H UniProt isoform 2/Q13507-2) is also located in a region of the channel involved in gating [44]. The arginine residue is located in the highly conserved TRP helix of the channel (Figure 1A), which connects the ICD of the channel to the S6 helix via a linker and is important for transducing conformational changes within the ICD to the channel gate [16].

Notably, the mGluR1-TRPC3 signalling pathway is dysregulated in a number of other SCAs, including two of the most common subtypes, SCA1 and SCA2 [48,49] (Figure 3). SCA44 is a spinocerebellar ataxia arising from mutations in mGluR1 receptors, which in turn may lead to aberrant Ca^2+^ signalling through TRPC3 [50]. Aberrant mGluR1-TRPC activity has also been described in mouse models of SCA3, SCA5, SCA14, SCA23 and SCA28 (reviewed in [51]). It is conceivable that the high expression of the more constitutively active splice variant TRPC3c in the cerebellum may exacerbate Ca^2+^ overload and contribute to the selective vulnerability of cerebellar Purkinje cells in SCAs. Inhibition of TRPC3 may therefore be a promising therapeutic strategy to treat multiple SCA subtypes.

Similar to its role in firing frequencies of cerebellar Purkinje cells, TRPC3 has been identified as an essential, but not sole, determinant of firing frequency in mouse midbrain dopaminergic (DA) neurons [52]. Midbrain DA neurons are robust pacemaking neurons that are not easily perturbed by pharmacological modulation, and their dysfunction is associated with Parkinson’s disease. Inhibition of TRPC3 channels with the pyrazole-based inhibitor Pyr10 completely abolishes spontaneous firing activity both in DA neurons in midbrain slices and in dissociated DA neurons. This inhibitory effect of Pyr10 on spontaneous firing is absent in *Trpc3*−/− mice. DA neurons recorded in midbrain slices from *Trpc3*−/− mice fire normally, however, due to compensatory mechanisms thought to involve the non-selective Na^+^ channel NALCN [52]. Due to its inward leak-type Na^+^ conductance at negative membrane potentials, TRPC3 strongly contributes to the slow depolarisation phase between action potentials in DA neurons. A similar role for TRPC3 has been demonstrated in mouse basal ganglia GABAergic projection neurons, whereby an inward Na^+^ leak-type conductance contributes to firing frequency and the characteristic depolarised membrane potential of these neurons [53]. TRPC3 is also expressed in cerebral vascular smooth muscle cells and in hippocampal neurons, and the channel has been implicated in epileptogenesis and status epilepticus. Inhibition of TRPC3 reduces pilocarpine-induced seizures in mice [54,55,56], which raises the possibility that TRPC3 inhibition could be a novel intervention for seizure disorders. 

In mouse trigeminal ganglia neurons TRPC3 mediates acute and chronic itch, and activation of the channel with GSK_170_ induces an acute itching behaviour, which is not seen in *Trpc3*−/− mice. Further, both genetic deletion and pharmacological inhibition of TRPC3 rescues inflammatory itch in a mouse model of contact hypersensitivity [57]. 

### 2.2. The Role of TRPC3 Outside of the Nervous System

TRPC3 is expressed in cardiovascular cells including cardiac fibroblasts, vascular smooth muscle cells and endothelial cells. As such, the channel has been identified as a potential player in, and therapeutic target for, cardiac pathologies including hypertrophy, atrial fibrillation and arrhythmias [32,33]. Ca^2+^ entry through TRPC3 has been linked to multiple downstream signalling pathways, such as activation of angiotensin II- and noradrenaline-induced nuclear factor of activated T cells (NFAT) [18]. Activation of the NFAT/calcineurin pathway via TRPC3 in cardiac myocytes is associated with pathologies such as arrhythmia and hypertrophy [32]. In addition, TRPC3 can form interactions with signalling partners such as the cardiac sodium-calcium exchanger NCX1 in cardiac myocytes [18,58,59,60], and aberrant TRPC3 activation is linked to arrhythmogenesis via Ca^2+^ overload and subsequent uncoupling from NCX1 [60]. Inhibition of TRPC3 has been explored as a possible intervention for cardiac hypertrophy in a mouse model, where some improvement was seen, but effects were limited by lack of TRPC3 inhibitor compounds with a desirable ADME (absorption, distribution, metabolism and excretion) profile [61]. TRPC3 has also been implicated in a number of cancers as a driver of tumour cell proliferation but targeting TRPC3 as a therapeutic strategy for cancer has not yet been explored (reviewed in [32]).

## 3. Regulation and Modulation of TRPC3

### 3.1. Endogenous Modulation of TRPC3

#### 3.1.1. TRPC3 Modulation by Lipids

The main activation pathway of TRPC3 and the closely related TRPC6 and TRPC7 channels is direct activation by DAG, downstream of GPCR activation. It has long been proposed that DAG interacts directly with the channel protein to induce activation [20], but until recently this had not been shown experimentally. Two lipid-binding fenestrations were identified in a cryo-EM structure of TRPC3; L1 residing in the VSLD, and L2 located in the pore-forming region of the channel [13] (Figure 1). Further functional studies using a photoswitchable DAG analogue as a probe (OptoDArG), and molecular dynamics (MD) simulations, have confirmed L2 as the main binding region for DAG [21,22,23]. Use of OptoDArG has also shown that partial lipidation of the closed state of the channel at subthreshold concentrations, is able to pre-sensitise the channel to subsequently open with faster activation kinetics when exposed to above-threshold concentrations [22]. Presumably, initial lipidation of the channel induces conformational changes in L2 that reduce the energetic barrier for subsequent channel opening transitions. 

A single residue, G652 (UniProt isoform 3/Q13507-3), within the L2 lipidation site and directly behind the selectivity filter has been identified in TRPC3 as being important for DAG binding and activation. Mutation of this glycine residue to alanine completely abolished DAG activation. The equivalent residue in the DAG-insensitive TRPV1 channel is a leucine, and mutation of the glycine to leucine in TRPC3 also abolished DAG activation [23]. The glycine residue is thought to act as a flexible linker that enables structural rearrangements within the pore-forming region following lipid activation [23,24]. Substitutions to bulkier amino acids which limit flexibility of the linker impair DAG activation in a graded manner [23]. In another study examining TRPC6, however, mutation of any of six residues within the same region abolished DAG activation [47]. It is therefore likely that there are additional important residues for coordinating DAG binding within the identified hydrophobic lateral fenestration in TRPC3. Interestingly, the DAG-insensitive G652A channel mutant retains its relatively high basal activity, suggesting that basal TRPC3 activation is DAG-independent [23].

In addition to regulation by DAG, other lipids present in the membrane environment have been found to regulate TRPC3 channel activity. For example, direct application of phosphatidylinositol 4,5-bisphosphate (PIP_2_), the precursor of DAG, has been shown to activate TRPC3 expressed in HEK293T cells in inside-out patch clamp experiments through undetermined mechanisms [62]. Conversely, PIP_2_ can also allosterically impair TRPC3 activation by GSK_170_, by binding L1 and regulating access of GSK_170_ to its binding site [63]. An increase in membrane cholesterol has also been found to potentiate TRPC3 activity, in part due to enhanced recruitment of the channel protein to the membrane [64]. In molecular dynamics simulations, when DAG is unbound from the two lipidation sites L1 and L2, other lipids such as cholesterol, phosphatidylcholine and phosphatidylserine occupy these sites, but the physiological relevance of this, if any, is unknown [22].

#### 3.1.2. Calcium Regulation of TRPC3

The activation of TRPC3 and closely related TRPC6/7 is transient, with responses to both endogenous and exogenous activators diminishing within seconds-to-minutes in whole-cell patch clamp recordings. It was previously not understood how this desensitisation may be arising, but it has been shown recently that increases in intracellular Ca^2+^ rapidly inhibit TRPC3/6/7 currents. As discussed, TRPC3 is a key regulator of Ca^2+^ homeostasis both in the brain and cell types outside of the CNS. As such, Ca^2+^-dependent inactivation of TRPC3 is likely to be an in-built mechanism for limiting excessive Ca^2+^ influx and toxicity. Proposed mechanisms for the regulation of TRPC3 by Ca^2+^ previously included CaM or CaMKII [65,66]. However, the recently resolved cryo-EM structures of both TRPC3 and TRPC6 in nominally Ca^2+^-free (1 mM EDTA) and high Ca^2+^ (1.34 µM) conditions coupled with electrophysiological characterisation, have led to the understanding that Ca^2+^ ions can directly modulate channel activity via distinct Ca^2+^ binding sites [15].

Previous cryo-EM structures of TRPC3 did not overlap well, but it is now thought that this is due to these being resolved in different Ca^2+^ conditions. TRPC3 structures can be grouped in terms of structural similarities depending on whether they were resolved in high Ca^2+^ concentrations [14,15,16] or low Ca^2+^ concentrations [13,15]. The disparities lie within the ICD of the channel, which is tightly packed and barely permeable to ions in Ca^2+^-inhibited conditions, and more loosely packed and conducting in low Ca^2+^ conditions.

In TRPC3, three putative Ca^2+^ binding sites have been identified, CBS1, 2 and 3 (Figure 1). CBS1 is formed at the interface between AR2 and the VH. CBS2 is located in the linker between the HH and VH, and CBS3 is located in the transmembrane region within the VSLD. In low Ca^2+^ conditions, it is proposed that negatively charged residues on CBS1 and CBS2 repel each other, dissociating AR2 and VH, and leading to the loosely packed ICD structure which is permeable to ions. Binding of a Ca^2+^ ion to CBS1 in high Ca^2+^ conditions neutralizes the negative charges between CBS1 and 2, stabilising the AR2 and VH interface and allowing the ICD to assume a compact, non-conducting conformation. Mutation of a critical Ca^2+^ binding residue within CBS1, D798, to alanine impaired Ca^2+^ inhibition of TRPC3 channels expressed in HEK293T cells in inside-out and whole-cell patch clamp experiments [15]. Mutation of E789, located in CBS2, to an alanine had no effect on Ca^2+^ inhibition, however, suggesting that CBS2 is less critical for inhibition of TRPC3 by Ca^2+^ ions. Ca^2+^-dependent inhibition at CBS1 and 2 is thought to arise from Ca^2+^ influx. Whole-cell TRPC3 currents are inhibited by high extracellular Ca^2+^ at negative membrane potentials, where ion flux is inwards according to the reversal potential for TRPC3 channels at physiological ion concentrations. No inhibition is seen at positive membrane potentials where ion flux is outwards.

Due to the location of CBS2 within the linker region between the HH and VH [15], it is possible that inhibitory Ca^2+^ binding effects are transduced to the transmembrane domains via the allosteric machinery identified by Sierra-Valdez and colleagues discussed earlier [16]. In agreement with this, structures resolved in low Ca^2+^ show upwards bending of the horizontal helix (Figure 2A–C), suggesting increased coupling between the ICD and the transmembrane domains [13,15]. Structures resolved in high Ca^2+^ have flatter horizontal helices [14,15,16] (Figure 2D,E). Additionally, the shortened HH and disrupted CIRB domain in the short channel isoform, TRPC3c, may impact on its Ca^2+^ regulation, which is deserving of further investigation.

Whilst the identified CBS are conserved between TRPC3 and TRPC6, at low concentrations of Ca^2+^, there is some degree of Ca^2+^ activation in TRPC6 channels. With increased Ca^2+^ concentrations this is overcome, and TRPC6 becomes Ca^2+^-inhibited similarly to TRPC3. Mutagenesis studies identified CBS3, located within the VSLD, as the activating Ca^2+^ binding site. Although binding of Ca^2+^ alone is not sufficient to evoke TRPC3 currents in the same manner as TRPC6, it is still possible that binding of Ca^2+^ ions within CBS3 is involved in the activation gating machinery of the channel, evidenced by dilation of the channel gate when Ca^2+^ ions are present in CBS3 [15]. Notably, CBS3 is a conserved cation coordination site in TRPC4/5 [67,68,69], and the equivalent residues form a Ca^2+^ activation site in TRPM2 channels [70].

A more recent study looked at the contribution of a hydrophobic tunnel formed by the symmetrical arrangement of VHs from the four subunits in TRPC3 tetramers to Ca^2+^ regulation. Four residues within the VH were identified which, when mutated, attenuated Ca^2+^ inhibition of receptor-activated whole-cell TRPC3 currents recorded from HEK293T cells [71]. This was in a graded manner correlating to hydrophobicity of the side chain substitution, and it was hypothesised that the hydrophobic tunnel influences CBS1 function. 

Interestingly, Ca^2+^-dependent inhibition of basal TRPC3 currents recorded in the inside-out configuration persists when the channel is co-expressed in HEK293T cells with an inactive CaM mutant [15], which originally lead to the conclusion that Ca^2+^ inhibition of TRPC3 is independent of CaM. However, recent whole-cell patch clamp experiments revealed that both co-expression of the inactive CaM construct and use of CaM antagonist calmidazolium diminished Ca^2+^ inhibition of receptor-activated TRPC3 currents [71]. It is possible that patch excision negates the need for CaM regulation, or that CaM is only involved in Ca^2+^ regulation downstream of receptor activation. Considering this role for CaM, it is likely that Ca^2+^ regulation is altered in TRPC3c, which has an impaired CIRB domain [16,27], and might explain the enhanced activity of this isoform.

### 3.2. TRPC3 Activators and Their Mode of Action

Most activators of TRPC3 identified to date do not discriminate between TRPC3/6/7 channels, due to their high sequence homology. In order to study the properties of TRPC channels in an isotype-specific manner, an activator which acts independently of PLC and DAG is highly desirable due to the promiscuity of this pathway. A number of synthetic activator compounds have been developed in recent years, some of which can discriminate between TRPC3/6 and 7, and act independently of DAG.

The benzimidazole GSK_170_ is the first and most widely reported synthetic small molecule activator of TRPC3 in the literature [60]. The agonist also activates TRPC6 and 7, however, so is not useful for distinguishing between the individual TRPC conductances in native cell types. A photoswitchable analogue of the compound has also been developed, OptoBI-1, which has been used for optical activation of TRPC3 channels in native cell types [72].

GSK_170_ reversibly enhances TRPC3 currents at sub-micromolar concentrations by interacting with residues located behind the selectivity filter that overlap with, but are not identical to, the DAG-interacting amino acids. Mutation of a critical glycine residue for lipidation by DAG and its analogues, G652, to alanine does not impair GSK_170_ activation of TRPC3 [23]. Rather, it enhances activation by GSK_170_ and its structural analogue BI-2, by stabilising the open conformation and inducing a prolonged single channel open state when the mutant channel is expressed in HEK293T cells [24]. Although the residue is not the binding site for GSK_170_, its flexibility and position behind the selectivity filter mean it is likely important in structural rearrangements of the pore following activation [13,23,24]. Mutation of F618, which resides within the same hydrophobic lateral fenestration as G652, to an alanine eliminates both DAG and GSK_170_ activation of the channel [23]. GSK_170_ is unable to activate channels which have become desensitised following lipid activation of TRPC3, and maximally lipid-activated channels cannot be further activated by GSK_170_ [24], again pointing towards shared activation pathways.

In silico docking of GSK_170_ and a structurally distinct agonist, M085, in TRPC6 combined with mutagenesis identified an extracellular membrane-facing binding pocket formed by the S6 helix and pore-forming loop of adjacent subunits [73]. A cryo-EM structure of TRPC6 in complex with another potent agonist, AM-0883, highlighted the same site [47]. Furthermore, activation by GSK_170_ only persists in outside-out excised patches, not inside-out patches, implying that the activator can only access its binding site extracellularly and has poor membrane permeability [73]. This pocket is proposed to overlap with the L2 DAG binding fenestration for TRPC6, similarly to TRPC3. All residues identified in this study are conserved in TRPC3 and as such the activation mechanisms are likely shared.

The short TRPC3c isoform is reportedly more sensitive to pharmacological potentiation by GSK_170_ [16]. It is unclear how this arises mechanistically, but a lowered energetic barrier for channel opening resulting from loss of exon 9, and the associated allosteric effects on the channel gate, may prime the channel for activation.

A medium-throughput screen of 2000 known compounds identified the anti-malarial compound artemisinin as being a selective activator of TRPC3 channels with an EC_50_ in HEK293 cells of 32.8 µM. The compound is thought to activate TRPC3 in a PLC- and DAG-independent manner and discriminates well between TRPC3, 6 and 7 [74]. Interestingly, co-application of artemisinin and synthetic DAG analogue, oleoyl-2-acetyl-sn-glycerol (OAG), shifts the EC_50_ for OAG activation to lower concentrations, suggesting that the anti-malarial compound can allosterically enhance OAG activation of the channel. Furthermore, OAG, GSK_170_ and other activators of TRPC3 all induce a rapid current rundown, thought to arise from Ca^2+^ inhibition of the channel [15]. Channel activation with artemisinin is sustained, however, with no apparent rundown [74], which implies that the ligand bypasses or impairs inhibitory regulation of the channel. Whether this is an allosteric effect or perhaps steric occlusion of the inhibitory Ca^2+^ binding sites is deserving of further investigation. This is seen for other TRP channel agonists, for example the TRPA1 activator GNE551. GNE551 binds within a pocket located at the interface between the S4 helix of one subunit and the pore-forming region of an adjacent subunit, and bypasses desensitization [75].

A high-throughput screen identified the pyrazolopyrimidine compound 4n, which activates TRPC3 currents at nanomolar concentrations and has 70-fold selectivity compared to TRPC6 channels [76]. No information regarding its mechanism of action is currently known, but close analogues of this compound were found to be inhibitors of TRPC6 channels rather than agonists [76].

### 3.3. TRPC3 Inhibitors and Their Mode of Action

Similar to channel agonists, the development of selective and potent inhibitors of TRPC channels has been a challenge due to their high sequence similarities (reviewed in [77]). Historically, pan-TRP channel modulators have often been used to investigate channel functions in native tissues. For example, 2-APB inhibits TRPC and TRPM channels, activates TRPV channels, and also inhibits IP3 receptors [28,78,79,80]. Phenylethylimidazole SKF96365 has been used as a TRPC channel inhibitor but is known to inhibit other Ca^2+^ permeable channels and even some K^+^ channels [81,82,83,84]. Recently, however, progress has been made on the development of more selective inhibitors and a number of potent TRPC3 inhibitors have been reported in the literature along with investigations into their mechanisms of action.

#### 3.3.1. Naturally Occurring TRPC3 Inhibitors

Both naturally occurring and synthetic steroids inhibit TRPC channels. Norgestimate inhibits TRPC3 and TRPC6 at low micromolar concentrations and inhibits other TRPC channels at higher concentrations [85]. Progesterone inhibits TRPC3 but is not selective for the TRPC3/6/7 subgroup and also inhibits TRPC1/4/5 [85,86].

A medium-throughput screen of plant-derived extracts identified compounds larixyl-acetate and larixol as inhibitors of the TRPC3/6/7 subgroup, but both inhibit TRPC6 most potently, in the submicromolar range [87]. The IC_50_ for inhibition of currents produced by co-expression of TRPC3 and TRPC6 in HEK293T cells, which would result in heteromeric co-assembly, was an intermediate between the IC_50_ values for the homomeric channels. This implies that the binding sites can act independently of the tetrameric assembly, with the composition of different binding sites affecting sensitivity to the antagonists.

#### 3.3.2. Pyrazole Compounds

Pyr3 was one of the first more selective TRPC3 inhibitors developed, which inhibits TRPC3 with submicromolar potency and does not inhibit TRPC6 or other TRPC channels. Photoaffinity labelling was used to demonstrate direct interaction of the inhibitor with the channel, and it was proposed that the compound binds to an extracellular site due to a lack of effect in inside-out patch clamp recordings in HEK293T cells [88]. However, it was later found that the compound inhibits store-operated Ca^2+^ entry processes via blockade of Orai1, with almost identical potency to its inhibition of TRPC3 [89]. Similarly, JW-65 from a series of pyrazole derivatives developed to improve upon metabolic properties and stability of the original compounds, inhibited STIM-Orai1 channels with similar potency [90]. This compound did successfully prevent initiation of pilocarpine-induced seizures in mice, however, demonstrating CNS efficacy [55].

#### 3.3.3. 2-Anilino-Thiazole Compounds

A series of compounds sharing a 2-anilino-thiazole core have been developed by GlaxoSmithKline (GSK) which inhibit TRPC3 and TRPC6 channels at low nanomolar concentrations and are selective against other TRP channels. GSK2833503A and GSK2332255B for example, inhibit TRPC3 and TRPC6 with IC_50_ values in the range of 3–21 nM in HEK293T cells [61,91]. Both compounds have poor bioavailability, however, and their use in vivo in mouse and rat models for cardiac hypertrophy was limited greatly by rapid metabolism and high protein binding [61].

BTDM is another compound from this potent series of 2-anilino-thiazole derivatives, which has equal potency for both TRPC3 and TRPC6 (IC_50_ values of 11 nM and 10 nM, respectively). A cryo-EM structure of this compound in complex with TRPC6 has been resolved [13]. The structure highlighted a binding pocket formed by residues on the S4–S5 linker on one channel subunit and residues on the S5 and S6 pore-forming helices of an adjacent subunit (Figure 4). The ligand is proposed to wedge between the pore-forming region and VSLD in the channel heterotetramer, hindering S4–S5 linker movement, and mutagenesis has confirmed the importance of these residues in channel inhibition [13]. All identified residues are conserved in TRPC3 channels, and since the compound is equipotent against both channels, the mechanisms are likely the same.

A later cryo-EM structure of TRPC6 identified a phosphatidylcholine lipid wedged between the S4 helix and S4–S5 linker [47], which is suggested to be an inhibitory lipid that limits S4–S5 linker flexibility. This is a phenomenon seen in TRPV1 and other TRPV channels with lipids resolved in the equivalent structural region, and these lipids can be displaced by agonist and antagonist binding [92,93]. The pocket occupied by the phosphatidylcholine lipid overlaps with the BTDM binding site, so it is possible that BTDM displaces a lipid in TRPC6 to exert its inhibitory action.

#### 3.3.4. Indane Derivatives

A series of indane derivatives have appeared in the literature, originally developed by Amgen, which have very high potency as TRPC3/6/7 channel inhibitors [47,94]. SAR-7334 for example, has nanomolar potency and is orally bioavailable [94]. Similarly, AM-1473, an analogue of SAR-7334, inhibits TRPC3 and TRPC6 with IC_50_ values of 8 nM and 0.22 nM, respectively, making it the most potent inhibitor in the literature to date [47].

A cryo-EM structure of TRPC6 in complex with SAR-7334 has been resolved, enabling the critical antagonist-binding residues to be identified [15] (Figure 4). The binding site is overlapping with CBS3, and it is proposed that part of the inhibitory mechanism of the compound in TRPC6 is occlusion of the activating Ca^2+^-binding site. An inhibitor of TRPC5, clemizole, was found to impair Ca^2+^ activation of this channel by binding the equivalent pocket to CBS3 in TRPC6/3 [69]. Since CBS3 was found to have less importance in regulation of TRPC3, this might explain the reduced potency of SAR-7334, and likely AM-1473, for inhibiting TRPC3 channels [15].

AM-1473 was found to bind to the same region as SAR-7334 in TRPC6 [47]. Notably, although the binding pockets in TRPC3 and TRPC6 are almost identical, one arginine in the TRPC6 pocket, R758 (Uniprot isoform 1/Q9Y210-1), is a lysine at the equivalent position in TRPC3, which could explain the difference in potency inhibiting these two channels. However, mutation of this lysine to arginine in TRPC3 did not fully restore the difference in potency between the two channels. Likewise, mutation of the arginine to lysine in TRPC6 only resulted in a five-fold reduction in potency for AM-1473 [47], suggesting that this residue alone is not accountable for the observed difference in potency.

In addition to impairment of CBS3 [15], it has also been suggested that the phosphatidylinositol lipid occupying the pocket near to the S4–S5 linker is involved. The lipid is present in the AM-1473-bound TRPC6 structure and may be stabilised within its binding site by AM-1473 to again limit S4–S5 linker movement and allosterically inhibit channel opening [47]. Further functional and structural studies will be required to confirm this.

## 4. Conclusions and Perspectives

Normal functioning of TRPC3 is important for both regulating Ca^2+^ homeostasis in cells throughout the body and regulating pacemaking in different neuronal subpopulations. The channel is potentially a useful therapeutic target for multiple pathologies. Since most TRPC3-related pathologies involve increased activity of the channel, a selective inhibitor of TRPC3 may be useful for treating aberrant Ca^2+^ signalling in multiple disorders. The recent availability of cryo-EM structures of TRPC3 and other closely related TRP channels has highlighted key regulatory sites and provided information about the conformational changes associated with channel gating. This information will be important for the future development of TRPC3 modulators through structure-guided docking of small molecules. Moreover, understanding the sequence of events that take place during TRPC3 activation and modulation by exogenous and endogenous factors will be essential for the development of new pharmacological agents acting on the channel. For example, lipidation sites which contribute to activation of the channel may be potential sites for inhibitors which could impair DAG activation of TRPC3, either by directly occluding L2 or by allosterically impairing activation via the L1 site in a similar nature to PIP_2_. Ca^2+^ binding sites could be modulated with small molecules either to activate or inhibit the channel. Molecules that target the HH and impair its flexibility could theoretically attenuate TRPC3 activity by reducing allosteric coupling between the HH and the transmembrane helices. A remaining challenge for TRPC3 drug development is its similarity to the TRPC6 channel, but identification of key regulatory sites that operate differently to TRPC6, such as differences in the Ca^2+^ regulatory mechanisms will assist in development of more selective modulators.

## Figures and Tables

**Figure 2 cells-12-02215-f002:**
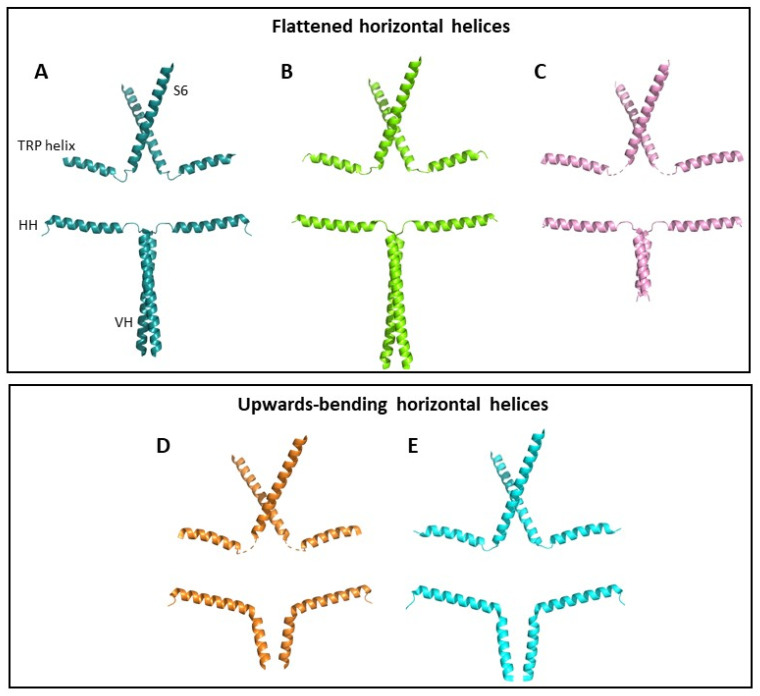
TRPC3 conformational states. Helical structures of two opposing TRPC3 subunits are shown from (**A**) PDB code 7DXB, (**B**) PDB code 5ZBG, (**C**) PDB code 6DJS, (**D**) PDB code 7DXC and (**E**) PDB code 6CUD. For each structure, S6 helix, TRP helix, horizontal helix (HH) and vertical helix (VH) are shown. Structures are grouped according to the position of their HHs, which are either horizontal (**A**–**C**) or upwards-bending (**D**,**E**). Other helices and subunits have been removed for clarity. Structures shown in (**A**–**C**) were resolved in high Ca^2+^ conditions, and structures (**D**,**E**) were resolved in low Ca^2+^ conditions.

**Figure 3 cells-12-02215-f003:**
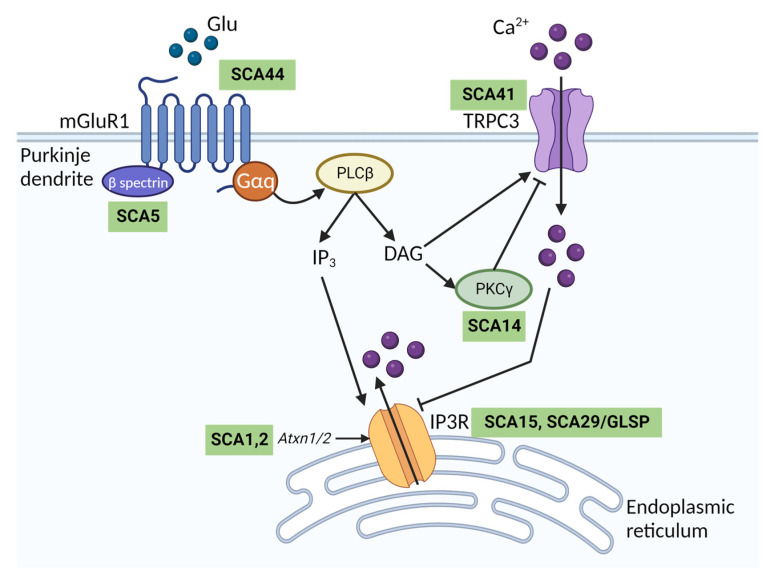
The mGluR1-TRPC3 signalling pathway in cerebellar Purkinje cells and spinocerebellar ataxia. Schematic diagram showing the downstream signalling cascade following metabotropic glutamate receptor type 1 (mGluR1) activation by glutamate (Glu) at Purkinje cell synapses. Glu released from granule cell parallel fibres binds mGluR1, which is coupled to Gαq, expressed on the Purkinje cell postsynaptic membrane. This activates phospholipase C beta (PLCβ) to produce inositol 1,4,5-trisphosphate (IP_3_) and diacylglycerol (DAG). IP_3_ activates inositol trisphosphate receptor (IP3R) in the endoplasmic reticulum to trigger Ca^2+^ release. DAG directly binds and activates TRPC3, resulting in Ca^2+^ influx. DAG and Ca^2+^ activate protein kinase C gamma (PKCƴ), which inhibits TRPC3 via phosphorylation. Human mutations in genes associated with different parts of the signalling cascade cause distinct spinocerebellar ataxia (SCA) subtypes as indicated in green boxes. SCA1 and 2, two of the most common SCAs, are caused by polyglutamine repeat expansions in *Atxn1* and *Atxn2* genes, respectively. The expanded genes have been associated both with reduced IP3R expression and increased sensitivity to activation by IP_3_ [41]. *GLSP* Gillespie syndrome. Figure created with BioRender.com.

**Figure 4 cells-12-02215-f004:**
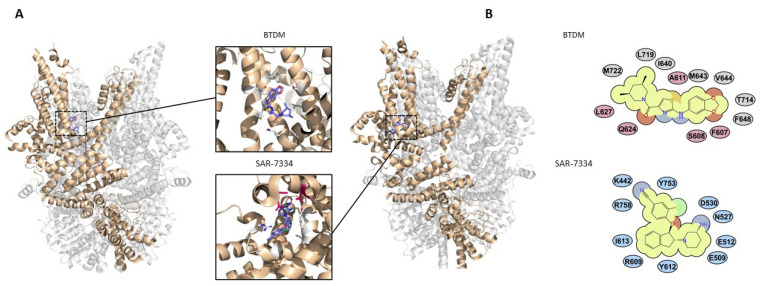
Binding sites of TRPC3/6 inhibitors. (**A**) BTDM (PDB 7DXG) and SAR-7334 (PDB 7DXF) binding sites in TRPC6 with interacting side chains shown. TRPC3/6 inhibitor AM-1473 binds within the same pocket as SAR-7334 [15,47]. CBS3 Ca^2+^ binding residues are highlighted in magenta. (**B**) Key BTDM- and SAR-7334-interacting residues in TRPC6 (adapted under CC-BY license from [15]).

## Data Availability

Protein structures used in this review are available from the Protein Data Bank (https://www.wwpdb.org/) accessed on 27 July 2023.

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
