# Peer review of "Modulation and Regulation of Canonical Transient Receptor Potential 3 (TRPC3) Channels"

_cells, 2023, doi:10.3390/cells12182215_

Round 1

Reviewer 1 Report

The authors reviewed the structure and function of TRPC3 Channels, and the role of TRPC3 in normal physiology and human diseases, as well as the modulators of TRPC3. It was well organized and discussed.

Minor:

1. The authors mentioned three isoforms of TRPC3, including TRPC3a, TRPC3b and TRPC3c, but just discussed the structures of TRPC3b and TRPC3c, no TRPC3a. The authors need to introduce something about TRPC3a or give the reason of no discussion of TRPC3a.

2. The authors can summarize the roles of TRPC3 on intracellular Ca2+ signaling and downstream effectors, not limit CaM.

Author Response

We thank the Reviewer for their critical insight and expertise, and for their comments that the manuscript is well-organised and discussed. 

We have considered all the comments and suggestions and have amended the manuscript accordingly.  We have also taken the opportunity to make minor edits to the text for accuracy, readability, and include additional insight from recent studies. 

The point-by-point response to each of the Reviewer’s comments and suggestions can be found below. 

Bethan A. Cole

Reviewer 1:

  1. The authors mentioned three isoforms of TRPC3, including TRPC3a, TRPC3b and TRPC3c, but just discussed the structures of TRPC3b and TRPC3c, no TRPC3a. The authors need to introduce something about TRPC3a or give the reason of no discussion of TRPC3a.

To date, no functional characterisation of TRPC3a has been carried out, so it is unclear how the longer isoform behaves compared to TRPC3b and c. We have added a sentence on line 92 to clarify this.

  1. The authors can summarize the roles of TRPC3 on intracellular Ca2+ signaling and downstream effectors, not limit CaM.

We have added more information about TRPC3 and other signalling cascades/ interacting proteins related to Ca2+ entry through TRPC3 throughout the review, for example CaN, NFAT and NCX1 in cardiac myocytes. We have also added additional information relating to TRPC3 regulation of IP3 receptors on line 163 of the revised manuscript.

Reviewer 2 Report

In the present review manuscript, the authors reviewed the canonical transient receptor potential 3 (TRPC3) channel role as a non-selective cation permeable channel with particularly high expression in the brain. The summarized its essential role in calcium signalling, and stated that the channel is activated directly by binding of diacylglycerol downstream of G-protein coupled receptor activation. In addition, TRPC3 is regulated by endogenous factors including Ca2+ ions, other endogenous lipids, and interacting proteins. Moreover, they review the recent literature which has advanced understanding of the complex mechanisms underlying modulation of TRPC3 by both endogenous and exogenous factors. TRPC3 plays an important role in Ca2+ homeostasis and entry into cells throughout the body, and both pathological variants and downstream dysregulation of TRPC3 channels have been associated with a number of diseases. Overall the manuscript is well written and nicely explained structure and function of the TRPC3. However, my suggestion for further improvement are as under

The lipid regulation is a general term mostly used in terms of lipid metabolism. However, here in these para it reflects the membran channels. I will recommend to change the title 3.1.1 with an appropriate tittle.

All functional roles of TRPC3 mentioned under different headings such as 2. The Role of TRPC3 in Normal Physiology and Human Disease, 3. Regulation and Modulation of TRPC3......and so on needs to be elaborated through gaphical presentations or may be summaized within tables.

Author Response

We thank the Reviewer for their critical insight and expertise, and for their comments that the manuscript is well-written and explained. 

We have considered all the comments and suggestions and have amended the manuscript accordingly.  We have also taken the opportunity to make minor edits to the text for accuracy, readability, and include additional insight from recent studies. 

The point-by-point response to each of the Reviewer’s comments and suggestions can be found below. 

Bethan A. Cole

Reviewer 2:

  1. The lipid regulation is a general term mostly used in terms of lipid metabolism. However, here in these para it reflects the membran channels. I will recommend to change the title 3.1.1 with an appropriate tittle.

As requested by the reviewer, we have changed the title of section 3.1.1 to ‘Modulation of TRPC3 by lipids’.

  1. All functional roles of TRPC3 mentioned under different headings such as 2. The Role of TRPC3 in Normal Physiology and Human Disease, 3. Regulation and Modulation of TRPC3......and so on needs to be elaborated through graphical presentations or may be summarized within tables

We have added a figure (Figure 3) depicting the mGluR1-TRPC3 signalling pathway and associated cerebellar diseases. Other roles outside of the cerebellum have not been elaborated in figures since evidence for these in humans is lacking in the current literature, which is now discussed in the text. Key regulatory sites related to section 3 are indicated in Figure 1 and 2 of the manuscript, so we felt that no further elaboration was needed for this.

Reviewer 3 Report

The manuscript by Cole and Becker reviews the literature describing the mechanisms of modulation of the TRPC3 channel. This is a timely and well-written review. There are, however, several statements that need to be revised in the review.
Major points:
1)    Abstract: TRPC3 channels are expressed not only in the brain, but also in some endocrine tissues and smooth muscle cells of the urinary bladder and blood vessels. The abstract should reflect that.
2)    The abstract contains two contradictory statements: 1. “The channel is activated directly by binding of diacylglycerol downstream of G-protein,” and 2. “The mechanisms underlying activation … are incompletely understood.” Please consider removing the second statement in the abstract or revising it.   
3)    Line 29: The sentence “The “canonical” or “classical” transient receptor potential (TRPC) subfamily consists of seven members, TRPC1-7, and has the closest sequence homology to the Drosophila TRP channel which was cloned originally [1,2]” needs to be revised. Currently, it appears that the cited references describe cloning of the Drosophila TRP channel. However, the papers described the cloning of mammalian TRPC1 channel. The history of TRPC channel cloning and discovery is reviewed in PMID: 32872338. This paper should be mentioned in the revised manuscript.
4)    Line 32: TRPC3 expression and functional characterization were first described in PMID: 9298988. This paper should be cited in the manuscript after “functionally characterised in vitro” on line 33. Currently, it appears that ref. 3 represents the first report describing expression and characterization of TRPC3, and this is not true. Ref. 3 describes the endogenous mRNA expression pattern of TRPC channels in the nervous system.
5)    Line 72: Please cite PMID: 9930701 here. This is the first report indicating that TRPC6 and TRPC3 are DAG-gated channels.
6)    Line 298: Please revise the following statement “suggesting that CBS2 is less critical for coordinating the inhibitory effect of Ca2+ ions.” It does not read right when you are talking about “coordinating the inhibitory effect of Ca2+ ions.”
7)    Line 330: Please revised the following statement “Interestingly, Ca2+-dependent inhibition of basal TRPC3 activity persists with co-expression of an inactive CaM mutant in excised inside-out patch clamp experiments [15]…” It does not sound correct when you write about co-expressing something in “excised inside-out patch clamp experiments.”
8)    The Perspectives section should be expanded.

Minor points:
1)    Line 48: Correct “ngeatively" to “negatively”
2)    Line 302: Please consider using “at” in the statement “…TRPC3 channels in physiological ion concentrations.”
3)    Line 443: Please insert a space between “21” and “nM.”
4)    Line 448: Please insert a space between “11” and “nM” as well as between “10” and “nM.”
5)    Line 474: Please insert a space between “8” and “nM” as well as between “0.22” and “nM.”
6)    Conclusions/ Line 499: Please consider removing the comma after “the body”

Author Response

We thank the Reviewer for their critical insight and expertise, and for their comments that the manuscript is well-written and timely. 

We have considered all the comments and suggestions and have amended the manuscript accordingly.  We have also taken the opportunity to make minor edits to the text for accuracy, readability, and include additional insight from recent studies. 

The point-by-point response to each of the Reviewer’s comments and suggestions can be found below. 

Bethan A. Cole

Reviewer 3:

  1. Abstract: TRPC3 channels are expressed not only in the brain, but also in some endocrine tissues and smooth muscle cells of the urinary bladder and blood vessels. The abstract should reflect that.

We have added a sentence on line 9 to indicate expression of TRPC3 in other cell types outside of the brain.

  1. The abstract contains two contradictory statements: 1. “The channel is activated directly by binding of diacylglycerol downstream of G-protein,” and 2. “The mechanisms underlying activation … are incompletely understood.” Please consider removing the second statement in the abstract or revising it.

We have amended the abstract in response to the reviewer’s comment. The second sentence has been changed to read “The molecular and structural mechanisms underlying activation and regulation of TRPC3 are incompletely understood.” to reflect current limitations in understanding with regards to downstream mechanisms as opposed to ligand activation of TRPC3.

  1. Line 29: The sentence “The “canonical” or “classical” transient receptor potential (TRPC) subfamily consists of seven members, TRPC1-7, and has the closest sequence homology to the Drosophila TRP channel which was cloned originally [1,2]” needs to be revised. Currently, it appears that the cited references describe cloning of the Drosophila TRP channel. However, the papers described the cloning of mammalian TRPC1 channel. The history of TRPC channel cloning and discovery is reviewed in PMID: 32872338. This paper should be mentioned in the revised manuscript.

We have included the suggested reference on line 31, which better describes the cloning of TRPC channels.

  1. Line 32: TRPC3 expression and functional characterization were first described in PMID: 9298988. This paper should be cited in the manuscript after “functionally characterised in vitro” on line 33. Currently, it appears that ref. 3 represents the first report describing expression and characterization of TRPC3, and this is not true. Ref. 3 describes the endogenous mRNA expression pattern of TRPC channels in the nervous system.

The suggested reference has been included in line 35.

  1. Line 72: Please cite PMID: 9930701 here. This is the first report indicating that TRPC6 and TRPC3 are DAG-gated channels.

The suggested citation has been added in on line 20.

  1. Line 298: Please revise the following statement “suggesting that CBS2 is less critical for coordinating the inhibitory effect of Ca2+ ions.” It does not read right when you are talking about “coordinating the inhibitory effect of Ca2+ ions.”

We have changed the sentence on line 329 of the revised manuscript to “…CBS2 is less critical for inhibition of TRPC3 by Ca2+ ions”.

  1. Line 330: Please revised the following statement “Interestingly, Ca2+-dependent inhibition of basal TRPC3 activity persists with co-expression of an inactive CaM mutant in excised inside-out patch clamp experiments [15]…” It does not sound correct when you write about co-expressing something in “excised inside-out patch clamp experiments.”

We have rephrased line 361 of the revised manuscript to “Interestingly, Ca2+-dependent inhibition of basal TRPC3 currents recorded in the inside-out configuration persists when the channel is co-expressed in HEK293T cells with an inactive CaM mutant”.

  1. The Perspectives section should be expanded.

We have expanded the Perspectives section in the revised manuscript to provide suggestions about future inhibitor modalities, and the future use of recent structural information for drug discovery.

  1. Line 48: Correct “ngeatively" to “negatively”

This typo has been corrected on line 50 of the revised manuscript.

  1. Line 302: Please consider using “at” in the statement “…TRPC3 channels in physiological ion concentrations.”

We have made the suggested change on line 333 of the revised manuscript.

  1. Line 443: Please insert a space between “21” and “nM.” Line 448: Please insert a space between “11” and “nM” as well as between “10” and “nM.” Line 474: Please insert a space between “8” and “nM” as well as between “0.22” and “nM.”

As suggested by the reviewer, we have added spaces for all units.

  1. Conclusions/ Line 499: Please consider removing the comma after “the body”

As suggested by the reviewer, we have removed the comma on line 532 of the revised manuscript.

Round 2

Reviewer 3 Report

This is a good review. It was a pleasure to read it. Great job!